# Oxygen isotopes in orangutan teeth reveal recent and ancient climate variation

**Tanya M Smith**[1,2]*, **Manish Arora**[3], **Christine Austin**[3], **Janaína Nunes Ávila**[1,4], **Mathieu Duval**[2,5,6], **Tze Tshen Lim**[7], **Philip J Piper**[8], **Petra Vaiglova**[1,2,8], **John de Vos**[9], **Ian S Williams**[10], **Jian-xin Zhao**[11], **Daniel R Green**[2,12]

[1]Griffith Centre for Social and Cultural Research, Griffith University, Southport, Australia; [2]Australian Research Centre for Human Evolution, Griffith University, Southport, Australia; [3]Department of Environmental Medicine and Public Health, Icahn School of Medicine at Mount Sinai, New York, United States; [4]School of the Environment, The University of Queensland, Brisbane, Australia; [5]Centro Nacional de Investigación sobre la Evolución Humana (CENIEH), Burgos, Spain; [6]Palaeoscience Labs, Department of Archaeology and History, La Trobe University, Melbourne, Australia; [7]Department of Geology, Universiti Malaya, Kuala Lumpur, Malaysia; [8]School of Archaeology and Anthropology, The Australian National University, Canberra, Australia; [9]Department of Geology, Naturalis Biodiversity Center, Leiden, Netherlands; [10]Research School of Earth Sciences, The Australian National University, Canberra, Australia; [11]Radiogenic Isotope Facility, School of the Environment, The University of Queensland, Brisbane, Australia; [12]Department of Human Evolutionary Biology, Harvard University, Cambridge, United States

*For correspondence:
tanya.smith@griffith.edu.au

**Competing interest:** The authors declare that no competing interests exist.

**Abstract** Studies of climate variation commonly rely on chemical and isotopic changes recorded in sequentially produced growth layers, such as in corals, shells, and tree rings, as well as in accretionary deposits—ice and sediment cores, and speleothems. Oxygen isotopic compositions ($\delta^{18}O$) of tooth enamel are a direct method of reconstructing environmental variation experienced by an individual animal. Here, we utilize long-forming orangutan dentitions (*Pongo* spp.) to probe recent and ancient rainfall trends on a weekly basis over ~3–11 years per individual. We first demonstrate the lack of any consistent isotopic enrichment effect during exclusive nursing, supporting the use of primate first molar teeth as environmental proxies. Comparisons of $\delta^{18}O$ values (n=2016) in twelve molars from six modern Bornean and Sumatran orangutans reveal a high degree of overlap, with more consistent annual and bimodal rainfall patterns in the Sumatran individuals. Comparisons with fossil orangutan $\delta^{18}O$ values (n=955 measurements from six molars) reveal similarities between modern and late Pleistocene fossil Sumatran individuals, but differences between modern and late Pleistocene/early Holocene Bornean orangutans. These suggest drier and more open environments with reduced monsoon intensity during this earlier period in northern Borneo, consistent with other Niah Caves studies and long-term speleothem $\delta^{18}O$ records in the broader region. This approach can be extended to test hypotheses about the paleoenvironments that early humans encountered in southeast Asia.

## eLife assessment

This **important** study presents **convincing** evidence for the use of orangutan teeth as terrestrial proxies to reconstruct rainfall regimes, while exploring the potentially conflicting impact of

breastfeeding signals. The findings will be of broad interest for those using and developing methods and tools to reconstruct environmental conditions in the historical and archaeological past.

## Introduction

Present-day rainfall patterns in Indonesia are controlled by the Asian and Australian monsoon systems, yielding annual trends that vary considerably with geography, topography, and the direction of monsoonal winds (*Aldrian and Dwi Susanto, 2003*; *Moron et al., 2009*; *Qian et al., 2013*; *Belgaman et al., 2017*). Northern Sumatra and western Borneo experience high annual rainfall and relatively stable annual temperatures, with a bimodal distribution of rainfall governed by the Intertropical Convergence Zone (*Van Schaik, 1986*; *Aldrian and Dwi Susanto, 2003*; *Belgaman et al., 2017*). These islands are also under the influence of inter-annual climate fluctuations driven by the El-Niño Southern Oscillation (ENSO); a periodic coupling of atmospheric and oceanic temperature gradients that initiates in the tropical Pacific, and influences global temperature and precipitation trends (*Marshall et al., 2009*).

It is well understood that variation in rainfall patterns influences the fundamental structure of primate habitats (*Brockman and Van Schaik, 2005*; *Wessling et al., 2018*). Dense tropical forests are sustained by fairly consistent rainfall and short, irregular dry seasons, while woodland communities in more arid environments have smaller trees, less dense canopies, and more deciduous trees (*Vico et al., 2017*; *Archibald et al., 2019*). In regions with prolonged dry seasons, low annual rainfall and savannah landscapes abound, in addition to disturbances such as wildfires (*Pletcher et al., 2022*).

Open woodland and savannah environments are unfavorable for slow-moving orangutans, the largest mammal with an arboreal lifestyle, particularly in regions with predators such as tigers or humans (*Thorpe and Crompton, 2009*; *Ashbury et al., 2015*; *Spehar et al., 2018*). Supra-annual ENSO events may also impact orangutan energy balance, reproduction, and social organization through the inducement of mast-fruiting, or dramatic seed production events in dipterocarp forests (*Knott, 1998*; *Curran et al., 1999*; *Marshall et al., 2009*). Such climate fluctuations over the past several hundred years have been documented in coral isotopes and tree-ring analyses, revealing especially marked changes during the past few decades (*Cole et al., 1993*; *Stahle et al., 1998*; *Hughen et al., 1999*; *Urban et al., 2000*; *Tudhope et al., 2001*; *Pumijumnong et al., 2020*).

Detailed climate records prior to the era of human-induced climate change are somewhat limited for island southeast Asia, but they are directly relevant to understanding the recent distribution of orangutans, and the arrival and dispersal of modern humans in the region during the Late Pleistocene (e.g. *Piper, 2016*; *Bae et al., 2017*; *Spehar et al., 2018*). A small number of studies of fossil corals, molluscs, marine sediments, and speleothems have provided insights into the last interglacial and glacial periods (e.g. *Hughen et al., 1999*; *Tudhope et al., 2001*; *Stephens et al., 2016*; *Yang et al., 2016*; *Buckingham et al., 2022*). For example, oxygen isotopes in fossil corals from seven periods during the last 130,000 years suggest that ENSO activity in the western Pacific over that time was comparable to modern records, although there was variation in the intensity of such activity at different timepoints (*Tudhope et al., 2001*). This study was also able to resolve bimodal annual rainfall peaks in modern corals, yet such detailed subannual records are extremely uncommon, particularly from terrestrial environments where early humans once lived alongside orangutans and other mammals.

## Oxygen isotope studies for paleoenvironmental reconstruction

Oxygen isotope values ($\delta^{18}O$) in water vary with latitude, altitude, temperature, and precipitation cycles, and are also impacted by precipitation sources. In tropical regions the primary determinant of rainfall isotope compositions is rainfall amount (*Dansgaard, 1964*; *Rozanski et al., 1993*; *Belgaman et al., 2017*). During wet seasons, rainfall $\delta^{18}O$ values are relatively low, while the opposite pattern is evident in periods with less rain, although other meteorological factors can influence isotope values as well (*Belgaman et al., 2016*). This primary tropical pattern influences isotopic variation in meteoric, surface, and leaf waters, which may show further elevations in $\delta^{18}O$ values during dryer periods due to preferential evaporative loss of the lighter isotope, $^{16}O$ (*da Silveira et al., 1989*; *Bowen, 2010*; *Roberts et al., 2017*).

**eLife digest** When an animal drinks water, two naturally occurring variants of oxygen – known as oxygen-18 and oxygen-16 – are incorporated into its growing teeth. The ratio of these variants in water changes with temperature, rainfall and other environmental conditions and therefore can provide a record of the climate during an animal's life. Teeth tend to be well preserved as fossils, which makes it possible to gain insights into this climate record even millions of years after an animal's death.

Orangutans are highly endangered great apes that today live in rainforests on the islands of Borneo and Sumatra. During a period of time known as the Pleistocene (around 2.6 million years to 12,000 years ago), these apes were more widely spread across Southeast Asia. Climate records from this area in the time before human-induced climate change are somewhat limited. Therefore, fossilized orangutan teeth offer a possible way to investigate past seasonal rainfall patterns and gain insight into the kind of environments early humans would have encountered.

To address this question, Smith et al. measured oxygen-18 and oxygen-16 variants in thin slices of modern-day orangutan teeth using a specialized analytical system. This established that the teeth showed seasonal patterns consistent with recent rainfall trends, and that the ratio of these oxygen variants did not appear to be impacted by milk intake in young orangutans. These findings indicated that the oxygen variants could be a useful proxy for predicting prehistoric weather patterns from orangutan teeth.

Further measurements of teeth from fossilized Sumatran orangutans showed broadly similar rainfall patterns to those of teeth from modern-day orangutans. On the other hand, fossilized teeth from Borneo suggested that the environment used to be drier, with less intense wet seasons.

The approach developed by Smith et al. provides an opportunity for scientists to leverage new fossil discoveries as well as existing collections to investigate past environments. This could allow future research into how climate variation may have influenced the spread of early humans through the region, as well as the evolution of orangutans and other endangered animals.

In addition to $\delta^{18}O$ values in fossil corals, tree rings, and speleothems, other fine-scaled oxygen isotopic climate proxies include otoliths (fish ear bones) and mollusc shells (e.g. *Aubert et al., 2012*; *Stephens et al., 2016*; *Prendergast et al., 2018*)—although these are rarely preserved in rainforest environments. Records of $\delta^{18}O$ values in mammalian tooth enamel are a more direct means of studying seasonality (reviewed in *Green et al., 2018*; *Green et al., 2022*), providing insight into the actual climates experienced by individuals, in contrast to indirect proxies for which it can be difficult to establish concurrence. Unlike bone, teeth do not remodel during life, and the phosphate component of the enamel mineral (hydroxyapatite) is especially resistant to modification after burial (reviewed in *Smith et al., 2018a*; *Pederzani and Britton, 2019*).

Tooth enamel is most commonly sampled with hand-held drills to recover the isotopic composition of oxygen inputs from water and food preserved in the hydroxyapatite (e.g. *Janssen et al., 2016*; *Roberts et al., 2020*; *Kubat et al., 2023*). This coarse drilling method yields spatially and temporally blurred powdered samples formed over a substantial and unknown period of time, however, precluding the identification of precise seasonal environmental patterns. To circumvent this limitation, we have employed the stable isotope sensitive high-resolution ion microprobe (SHRIMP SI) to measure $\delta^{18}O$ values sequentially from thin sections of teeth, relating these to daily increments and birth lines to determine enamel formation times, and in some instances, calendar ages (*Smith et al., 2018a*; *Smith et al., 2022*; *Green et al., 2022*; *Vaiglova et al., 2024*).

It is well established that $\delta^{18}O$ values in tooth enamel are closely related to local water oxygen isotope compositions (reviewed in *Green et al., 2018*; *Green et al., 2022*). For teeth that form after birth and during periods of milk consumption, $\delta^{18}O$ values are expected to be higher, as a result of infant evaporative water loss while consuming $^{18}O$-enriched mother's milk (*Bryant et al., 1996*; *Wright and Schwarcz, 1999*; *Britton et al., 2015*). Studies of large-bodied mammals report that milk $\delta^{18}O$ values are elevated by ~1–6‰ relative to local drinking water $\delta^{18}O$ (*Kornexl et al., 1997*; *Lin et al., 2006*; *Chesson et al., 2010*; *Green et al., 2018*; but see *Cherney et al., 2021*). Comparable data on human or nonhuman primate milk enrichment appear to be lacking, save for a study of

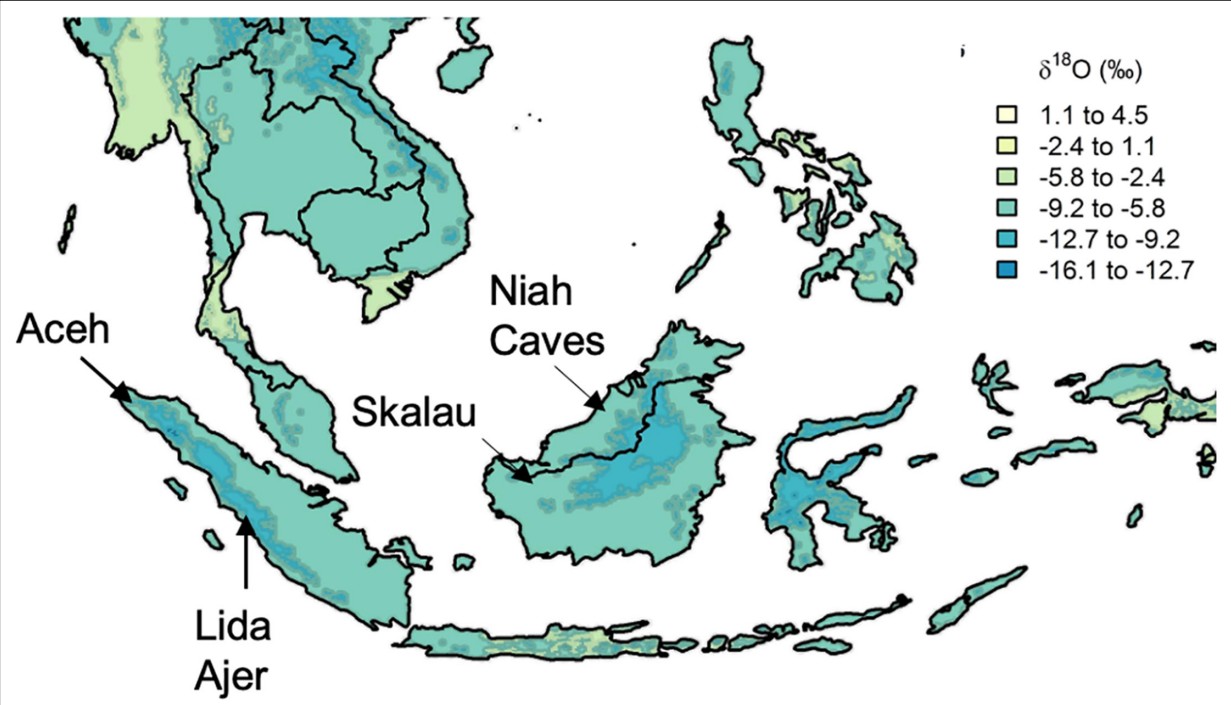

**Figure 1.** Approximate location of select modern and fossil orangutans superimposed on modeled isotopic variation. Figure modified from https://www.waterisotopes.org based on data from the Online Isotopes in Precipitation Calculator (3.0). See *Table 1* for the location of particular individuals. Sibrambang Cave has yet to be relocated since Eugene Dubois' original excavations, but it is known to be in the general vicinity of Lida Ajer in the Padang Highlands, possibly near to the modern village of a similar name (*Louys et al., 2024*).

44 British infants aged 5–16 weeks (*Roberts et al., 1988*). The urine of infants who were breast-fed showed isotopic enrichment of 1–3‰ compared to infants who were fed formula prepared from sterile local tap water.

While such studies point to potential changes in infant body water during nursing, it is unclear whether such differences prohibit the use of early-formed enamel in studies of climate variation (*Blumenthal et al., 2017*; *Luyt and Sealy, 2018*). Two studies of $\delta^{18}O$ values in the dentitions of modern sheep, horses, and zebras reported higher bulk values (~1–2‰) in five molars (M1) compared to the rest of the permanent dentition (*Bryant et al., 1996*; *Fricke and O'Neil, 1996*). This led *Fricke and O'Neil, 1996*, to suggest that M1s are unlikely to reflect the values of local meteoric water due to the influence of maternal inputs in utero and through lactation. However, near-weekly $\delta^{18}O$ values over the first 2.75 years of life in a Neanderthal M1 measured with SHRIMP SI showed clear annual trends and maximum $\delta^{18}O$ values corresponding to a period after nursing has ceased (*Smith et al., 2018a*). An examination of longer continuous periods of enamel formation within and between teeth will help to clarify whether early-formed primate teeth should be avoided for studies of climate seasonality.

Here, we first assess whether wild orangutans show elevated $\delta^{18}O$ values in early-formed enamel, testing the suggestion that M1s are significantly affected by nursing $^{18}O$-enrichment, thereby precluding their use in climatological reconstructions. We then explore approximately 30 years of weekly $\delta^{18}O$ values (n=2016 measurements) to compare orangutan individuals from the islands of Sumatra and Borneo. Finally, we contrast $\delta^{18}O$ values between modern and Pleistocene orangutans, including those from key regions of early human occupation: Lida Ajer, Sumatra (*Hooijer, 1948*; *Westaway et al., 2017*) and Niah Caves, Malaysia (*Hooijer, 1961*; *Barker et al., 2007*; *Figure 1*, *Table 1*). Novel understanding of climate patterns in these fossil assemblages may inform debates about the likelihood of modern humans living in dense Asian rainforests, and the conditions that would support savannah corridors for human dispersals throughout the region (e.g. *de Vos, 1983*; *Bird et al., 2005*; *Westaway et al., 2017*; *Louys and Roberts, 2020*; *Ao et al., 2024*; *Hamilton et al., 2024*).

**Table 1.** Modern and fossil orangutan teeth employed in the current study.

| Taxon | Accession | Origin | Sex | Age (years) | Teeth |
|-------|-----------|--------|-----|-------------|-------|
| *Pongo pygmaeus* | ZSM 1981/48 | Skalau, Borneo | F | ~8.4 | RUM1, LLM2 |
| | ZSM 1981/87 | Skalau, Borneo | F | >9 | LUM1, RUM2, RLM3 |
| | MCZ 5290 | Borneo (location unspecified) | n/a | 4.5 | RUM1 |
| *Pongo abelii* | ZSM 1981/246 | Aceh, Sumatra | M | ~8.5 | LLM1, LUM2 |
| | ZSM 1981/248 | Aceh, Sumatra | F | adult | LUM1, LUM2, LLM3 |
| | ZMB 83508 | Sumatra (location unspecified) | n/a | 8.8 | RLM1 |
| Fossil *Pongo* spp. | 11564.5 | Sibrambang, Sumatra | n/a | n/a | RUM |
| | 11565.162 | Sibrambang, Sumatra | n/a | n/a | LUM |
| | 11594.12 | Lida Ajer, Sumatra | n/a | n/a | RLM |
| | 11595.105 | Lida Ajer, Sumatra | n/a | n/a | LLM |
| | US/22 | Niah Caves, Malaysia | n/a | n/a | RLM |
| | Y/F4 | Niah Caves, Malaysia | n/a | n/a | LLM |

Numerous taxonomic assignments have been made for fossil orangutans (*Pongo* spp.), some of which have not been based on clear morphological characteristics (*Tshen, 2016*), and are not relevant for the focus of this paper.

## Results

### Modern orangutans

The $\delta^{18}$O ranges of twelve modern and six fossil orangutan molars, representing 2971 near-weekly measurements spanning 57.6 years of tooth formation, are listed in *Table 2*. Prior to making comparisons between individuals, geographic regions, or time periods, we first consider the potential intra-individual effect of isotopic enrichment from maternal milk on $\delta^{18}$O values. Comparisons of $\delta^{18}$O values during the first, second, and third years of life in five modern orangutan first molars (M1) do not show consistently elevated values during their first year (*Figure 2*). Mean yearly $\delta^{18}$O values in the first year are elevated by only 0.3‰ compared to the second year. While three of the five M1s showed first year $\delta^{18}$O values higher than second year values (p≤0.05), only two individuals showed mean values that were ~1–2‰ higher during year 1; one individual showed no difference from the first to the second year, and one individual showed lower values during the first year than during the second year (p≤0.05) (*Table 3*). A sixth individual was only sampled from 193 days of age, but maximum values from this point onward were similar across more than 3 years of life. Similarly variable patterns were observed for the six putative fossil orangutan M1s (*Appendix 1—figure 1*).

Comparisons across serial molars in four modern orangutans show no consistent trend of elevated $\delta^{18}$O values in M1s relative to successive molars (*Figure 3*). Only two individuals showed maximum $\delta^{18}$O values in their M1s relative to M2s; in both instances M3s were unavailable due to their lack of development prior to death. The other two individuals showed higher $\delta^{18}$O values in M2s or M3s than in their respective M1s. In the case of the oldest individual (ZSM 1981/248), the highest $\delta^{18}$O values appeared at approximately 5.8 years of age, well past the age when exclusive nursing ends.

Comparison of the $\delta^{18}$O values in the full datasets of modern Bornean and Sumatran orangutans reveals a high degree of overlap. Values from the three Bornean individuals ranged from 12.7‰ to 20.0‰ (n=955 near weekly measurements), while the three Sumatran individuals ranged from 11.3‰ to 20.6‰ (n=1061 measurements). Comparisons of periodic trends via spectral power distribution analysis revealed more consistent bimodal patterns in the Sumatran individuals; three of the six Bornean molars were aperiodic (statistical power of 0.1 or less), while all six of the Sumatran molars revealed annual or semiannual cycles with greater power (*Appendix 1—figure 2*). Rapid oxygen isotopic shifts on the order of ~6–8‰ are evident in the single Bornean and Sumatran individuals with $\delta^{18}$O measurements spanning M1 to M3, which may represent one or more supra-annual ENSO events captured during the ~9–11 years these molars were forming.

**Table 2.** Modern and fossil orangutan molar $\delta^{18}O$ values.

| Taxon | Accession | Tooth | Cusp | Spots | Time (days) | dO18 range |
|---|---|---|---|---|---|---|
| *P. pygmaeus* | ZSM 1981/48 | RUM1 | dl | 151 | 1241 | 13.6–19.9 |
| | ZSM 1981/48 | LLM2 | mb | 107 | 804 | 13.0–18.8 |
| | ZSM 1981/87 | LUM1 | ml | 131 | 869 | 13.7-17.9 |
| | ZSM 1981/87 | RUM2 | ml | 196 | 1195 | 12.7–20.0 |
| | ZSM 1981/87 | RLM3 | mb | 220 | 1350 | 13.7–19.2 |
| | MCZ 5290 | RUM1 | ml | 150 | 1002 | 13.8–18.1 |
| *P. abelii* | ZSM 1981/246 | LLM1 | mb | 136 | 1425 | 12.3–18.3 |
| | ZSM 1981/246 | LUM2 | ml | 229 | 1376 | 12.6–18.0 |
| | ZSM 1981/248 | LUM1 | db | 177 | 1072 | 11.3–19.3 |
| | ZSM 1981/248 | LUM2 | db | 193 | 1374 | 13.5–20.6 |
| | ZSM 1981/248 | LLM3 | db | 191 | 1461 | 14.8–19.6 |
| | ZMB 83508 | RLM1 | db | 135 | 1029 | 13.4–20.4 |
| Fossil *Pongo* spp. | 11564.5 | RUM | mb | 178 | 1387 | 15.3–20.4 |
| | 11565.162 | LUM | ml | 143 | 1144 | 14.7–20.8 |
| | 11594.12 | RLM | ml | 154 | 1081 | 15.1–19.9 |
| | 11595.105 | LLM | mb | 197 | 1312 | 15.7–20.0 |
| | US/22 | RLM | mb | 149 | 1023 | 15.9–24.8 |
| | Y/F4 | LLM | db | 134 | 869 | 14.2–22.9 |

## Fossil orangutans—oxygen isotopes

Concurrently forming teeth (molar specimens 11594.12 and 11595.105) from same individual at Lida Ajer, Sumatra, are nearly isotopically identical; $\delta^{18}O$ values range from 15.1‰ to 19.9‰ and 15.7‰ to 20.0‰, respectively, supporting the biogenic fidelity of these records. The $\delta^{18}O$ values of two individuals from the nearby Sibrambang site (15.3–20.4‰, 14.7–20.8‰) are very similar to those of the Lida Ajer individual. These Sumatran fossils all fall at the upper end of the range of modern Sumatran orangutans (*Figure 4*), and reveal approximately annual $\delta^{18}O$ periodicities (0.9–1.3 years), as well as strong bimodal distribution patterns in one instance (11565.162).

The two fossils from the Niah Caves were excavated from different regions and stratigraphic depths; $\delta^{18}O$ values in the tooth from grid US/22 ranged from 15.9‰ to 24.8‰ and, unlike the three modern Bornean individuals, yielded an annual periodicity (1.0 years). The $\delta^{18}O$ in the tooth from grid Y/F4 ranged from 14.2‰ to 22.9‰ and showed a stronger bimodal trend than an annual one, although its short formation time may have prohibited identification of longer trends. The range of values from these two fossil molars (14.2–24.8‰) markedly exceeds the range of modern Bornean orangutans (12.7–20.0‰) (*Figure 4*), with the mean $\delta^{18}O$ value at least 2‰ heavier. This suggests possibly drier conditions with greater seasonality during fossil molar formation (*Figure 4—figure supplement 1*).

## Fossil orangutans—U-series age estimates

The six fossil teeth have very low uranium concentrations in their enamel (<0.5 ppm), regardless of their origin (*Supplementary file 1*). These enamel values are very close to the detection limit of the Nu Plasma II MC-ICP-MS, and thus are not useful for estimating minimum ages. The dentine of Lida Ajer specimen 11595.105 shows a spatial gradient of increasing uranium concentration from ~41 to 66 ppm, and decreasing age estimates from ~51 to 40 ka (*Supplementary file 1*). This trend might result from a preferential uranium leaching overprint near the end of the root. Spot DE10, positioned near the enamel-dentine junction (EDJ), is less likely to be impacted (*Appendix 1—figure 3*), and is thus assumed to provide the most reliable minimum age for the tooth, ~40 ka. Uranium values from Lida Ajer specimen 11594.12 show a similar trend of concentrations decreasing from ~31 to 24 ppm

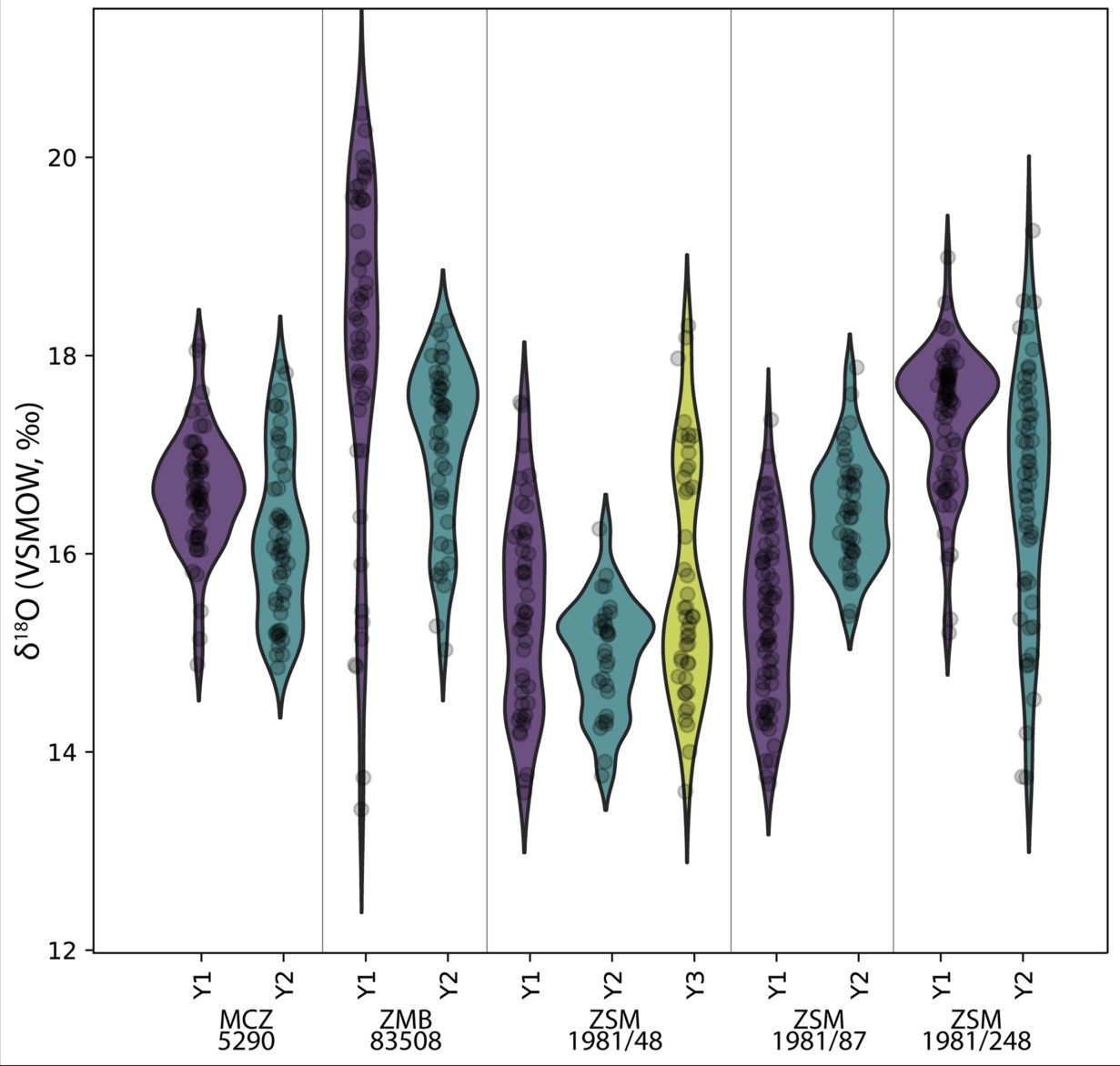

**Figure 2.** Comparison of sequential δ¹⁸O values across multiple years of first molar formation in five modern orangutans from Borneo and Sumatra. Bornean individuals: MCZ 5290, ZSM 1981/48, ZSM 1981/87; Sumatran individuals: ZMB 83508, ZSM 1981/248. The width of each curve is a kernel density estimate (KDE) corresponding to the distribution of δ¹⁸O values. First year data (Y1) is shown with a purple violin plot, second year data (Y2) with a green plot, and third year data (Y3) with a yellow plot where complete/available. Actual data are plotted as black circles.

toward the root tip. However, the U-series age estimates remain constant within the range 31–34 ka across the dentine (*Supplementary file 1*; *Appendix 1—figure 3*). No evidence for a recent overprint is observed, supporting a minimum age of 33 ka. In summary, this individual's age is at least 33 ka, and possibly >40 ka.

U-series analysis of the dentine of Sibrambang specimen 11565.162 shows a slight decreasing trend of uranium concentration from the EDJ to the root tip (from >60 ppm to <60 ppm), and corresponding increasing age estimates (56–62 ka) (*Supplementary file 1*; *Appendix 1—figure 4*). This might result from a slight uranium leaching overprint; a minimum age of 60 ka is likely for this tooth. The U-series age estimates obtained for Sibrambang specimen 11564.5 show a decreasing trend from the EDJ toward the circumpulpal dentine from 75 to 65 ka (*Supplementary file 1*; *Appendix 1—figure 4*). However, given the associated uncertainties, this trend might not be meaningful. An average dentine U-series age of 70.3±5.5 ka (2σ) may be regarded as a minimum age for the fossil, which is broadly

**Table 3.** Comparisons of first and second year $\delta^{18}O$ values in five first molars.

| Specimen | Adjusted p-values | Higher $\delta^{18}O$ values |
| --- | --- | --- |
| MCZ 5290 | p=0.010 | Year 1 |
| ZMB 83508 | p=0.006 | Year 1 |
| ZSM 1981/48 | p=0.161 (NS) | Year 1 |
| ZSM 1981/87 | p<0.001 | Year 2 |
| ZSM 1981/248 | p<0.001 | Year 1 |

consistent with the single age estimate obtained from the enamel (64 ka). In summary, the two teeth from Sibrambang yield U-series apparent ages of ~60–70 ka.

The uranium concentration measured across the dentine of the Niah Caves specimen from grid Y/F4 shows little variability, 4.2–4.9 ppm. The U-series age estimates are between 6.0 and 8.7 ka (*Supplementary file 1*; *Appendix 1—figure 5*). The average dentine U-series minimum age is 7.6±1.3 ka. Similarly, the Niah Cave specimen from grid US/22 shows a consistent uranium concentration through

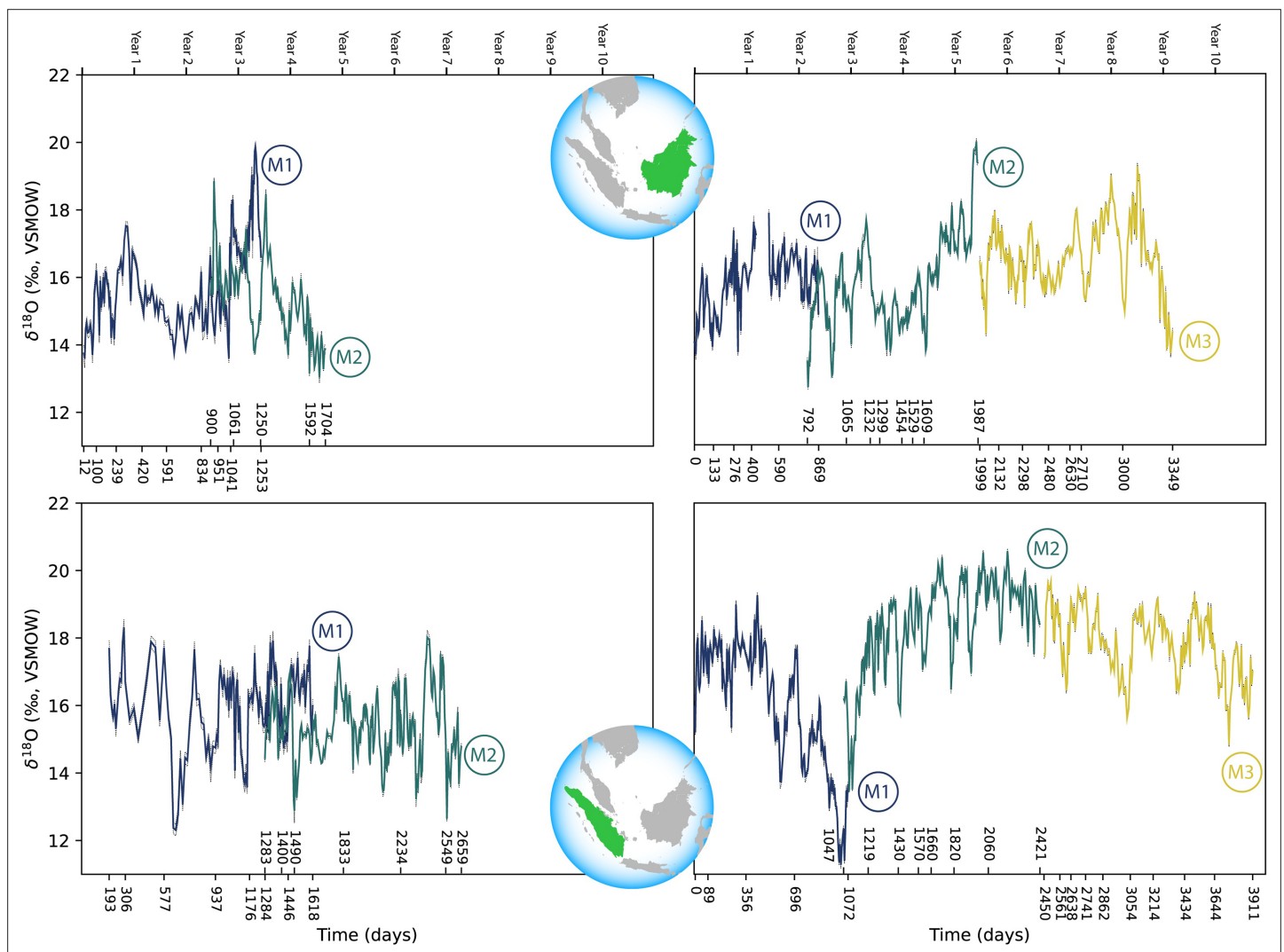

**Figure 3.** Comparison of sequential $\delta^{18}O$ values across multiple years of serial molar formation in two modern orangutans from Borneo (top) and two from Sumatra (bottom). Individual in upper left: ZSM 1981/48; upper right: ZSM 1981/87; lower left: ZSM 1981/246; lower right: ZSM 1981/248. Developmental overlap was determined through registration of trace elements as in *Smith et al., 2017*.

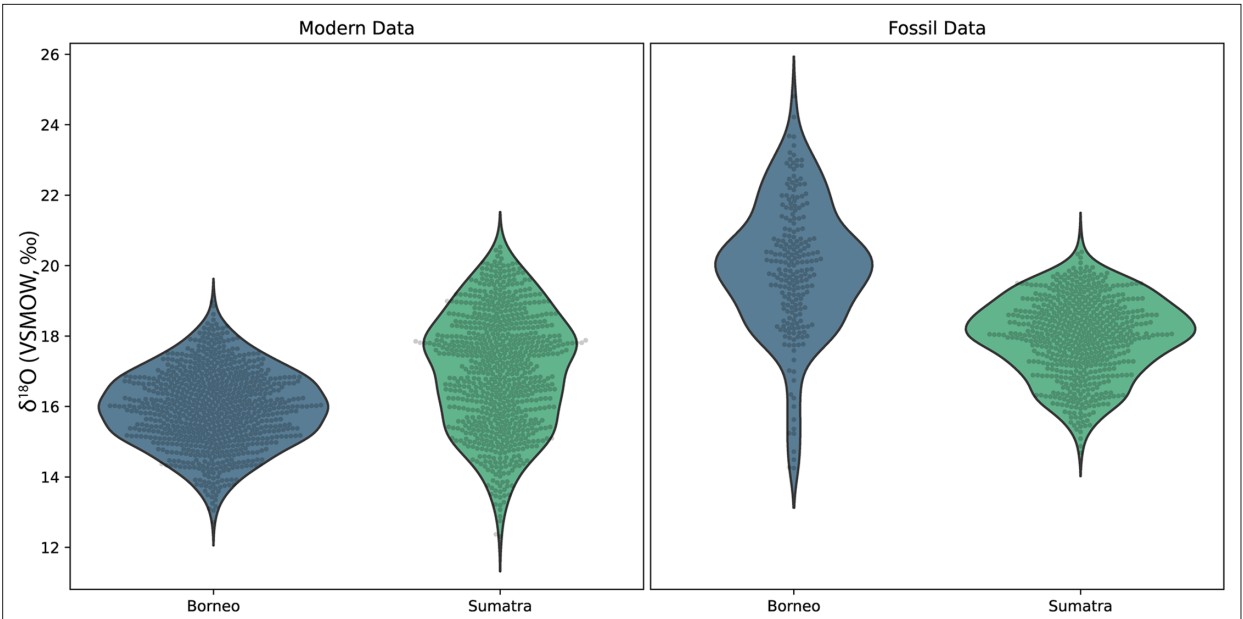

**Figure 4.** Comparison of δ¹⁸O values in fossil and modern orangutans from Borneo (blue) and Sumatra (green). Violin plots show kernel density estimates representing the distribution of δ¹⁸O values in modern individuals (left plot) and in fossil individuals (right plot). Actual δ¹⁸O measurements are shown as black circles.

The online version of this article includes the following source data and figure supplement(s) for figure 4:

**Figure supplement 1.** Comparison of δ¹⁸O values in fossil and modern orangutans from Borneo and Sumatra.

**Figure supplement 1—source data 1.** Numerical data used to generate this figure.

the dentine (1.3–1.4 ppm), with relatively large uncertainties that nonetheless bracket individual age estimates (*Supplementary file 1*; *Appendix 1—figure 5*). The average dentine U-series age is 8.8±3.0 ka. In summary, the two teeth from Niah Cave yield consistent apparent ages of ~8–9 ka, which should be regarded as a minimum age constraint for the fossils.

## Discussion
### Primate oxygen isotope compositions do not reveal a clear milk enrichment effect

Half of our modern sample, and potentially all of our fossil sample, are composed of M1s. These begin forming around birth and continue growing for 3 or more years (*Smith, 2016*). Orangutan infants rely exclusively on maternal milk during their first year of life, supplementing this with solid foods in the second year, which are increased until suckling ceases prior to 9 years of age (*van Noordwijk et al., 2013*; *Smith et al., 2017*). Our developmentally guided sampling approach allows us to examine fine-scaled trends in δ¹⁸O values during birth, exclusive nursing, supplemental feeding, and also after nursing ends (in those individuals with available serial molar teeth).

We find that five modern orangutans show only minor and inconsistently elevated δ¹⁸O values during the first year of life when compared to the subsequent year. These data do not support the hypothesis that primate infants have markedly elevated body water δ¹⁸O values during exclusive nursing. Data from the majority of 12 human M1s studied by *Vaiglova et al., 2024*, similarly reveal maximum δ¹⁸O values after the first year of tooth formation, well beyond the duration of exclusive milk intake. This is also evident in the M1 of a Neanderthal born in the spring (*Smith et al., 2018a*); δ¹⁸O values mostly rose for the first 3.5 months of life, but did not reach a maximum for another 2 years—long after the infant would have begun consuming supplemental foods and liquids. This final dataset points to the influence of season of birth on initial postnatal δ¹⁸O values, as inferred in other mammals (*Bryant et al., 1996*; *Fricke and O'Neil, 1996*).

Comparisons of serially forming teeth in four wild orangutans also fail to show a consistent elevation of δ18O values in M1s versus M2s (or M3s in two cases). Comparisons of M1 δ18O values with subsequent-forming teeth in four baboons, two tantalus monkeys, and one mona monkey (from *Green et al., 2022*: SI Dataset S1) also largely fail to support the enriched 'Pattern 1' trend modeled by *Bryant et al., 1996*: *Figure 4*, p. 401. This is also the case in comparisons of δ18O values from bulk samples of human teeth—*Wright and Schwarcz, 1999*, demonstrated that M1s have higher δ18O values than later-forming teeth in only four of seven individuals. In summary, the data from a range of primates including humans do not support the exclusion of early-forming primate teeth from the assessment of environmental seasonality.

## Modern orangutans show similar isotopic values across the islands of Borneo and Sumatra

The two Bornean juveniles from the Munich collection (ZSM 1981/48, ZSM 1981/87) reflect the environmental conditions of the late 1880s and early 1890s in Skalau—a region where orangutans might now be locally extinct. Similarly, the teeth from the two Sumatran individuals from the Munich collection (ZSM 1981/246, ZSM 1981/248) were collected prior to 1939 in northern Aceh, from where orangutans also have since disappeared (*Spehar et al., 2018*). While the individuals from northernmost Sumatra might have inhabited somewhat higher elevations than those from western Borneo, there does not appear to be an evident altitude effect (lower isotopic values at higher altitudes), as these four individuals show similar isotopic values, save for a single brief excursion below 12‰ in ZSM 1981/248 (*Table 1*, *Figure 3*). It is unknown to what extent local rainfall may have been isotopically distinct at the time the teeth were forming.

The δ18O values shown in *Figure 1* reflect estimates of monthly and annual average precipitation from the Online Isotopes in Precipitation Calculator (3.0) compiled for https://www.waterisotopes.org. Actual measurements of precipitation δ18O from the islands of Borneo and Sumatra are extremely limited. The closest observation facilities to the ZSM orangutan locations yield similar patterns of modern annual rainfall δ18O variability (*Belgaman et al., 2017*), yet specific measurements from the six facilities that make up 'Cluster 3' in this reference are not available for comparison.

Other studies underscore the complexity of water transport in this region—multiple factors such as the oceanic origin of water vapor, cloud cover and type, and the post-condensation process influence the short-term variability of δ18O values in rainfall (*Moerman et al., 2013*; *Suwarman et al., 2013*; *Belgaman et al., 2016*). For example, *Moerman et al., 2013*, provided 5 years of daily rainfall δ18O measurements from Northern Borneo (Gunung Mulu National Park, Malaysia); daily rainfall δ18O values ranged from +0.7‰ to −18.5‰ and showed 1–3 month, annual, and supra-annual cycle frequencies. Interannual rainfall δ18O fluctuations of 6–8‰ were significantly correlated with ENSO events; these are similar in scale to the large fluctuations in our serial tooth datasets (*Figure 3*).

Another potential source of isotopic variability derives from dietary variation, as orangutans obtain the majority of their body water from plants (*Mackinnon, 1974*). Plant oxygen isotope compositions can be stratified within tropical forest canopies (*da Silveira et al., 1989*; *Roberts et al., 2017*; *Lowry et al., 2021*)—potentially leading to offset values among various animals, including primates, that consume different resources in the same forest (*Krigbaum et al., 2013*; *Nelson, 2013*; *Fannin and Scott McGraw, 2020*). Orangutans forage at different canopy heights ranging from the ground to high in the canopy (*Mackinnon, 1974*; *Ungar, 1996*; *Thorpe and Crompton, 2005*; *Ashbury et al., 2015*). *Mackinnon, 1974*, reported that Bornean and Sumatran orangutans obtain 95% of their food from the middle and upper levels of the canopy, where preferred foods are most abundant. In contrast, *Ungar, 1996*, reported that Sumatran orangutans were quite variable in feeding heights, with a mean of approximately 19 m; lower than gibbons who fed preferentially in the high canopy. *Thorpe and Crompton, 2005*, reported stratification in Sumatran orangutans, with immature individuals feeding below 20 m, females feeding both below and above this height, and adult/subadult males preferring to feed high in the canopy.

While differences in enamel δ18O values are apparent in comparisons of sympatric arboreal and terrestrial mammals (reviewed in *Lowry et al., 2021*; *Green et al., 2022*), it remains to be seen whether primates with broadly similar diets and habitats show meaningful differences in δ18O values, and to what degree plant physiology influences the pattern and amplitude of seasonality relative to rainfall. Oxygen isotope compositions in the six modern individuals from the islands of Borneo and

Sumatra are very similar. Orangutans from both islands prefer ripe fruit when available, with some differences in the consumption of bark, leaves, unripe fruits, and insects—which varies between sites and across seasons (reviewed in *Smith et al., 2012*). Seasonal variation in diets and the stratification of food within the canopy may also contribute to enamel oxygen isotope variation within individuals, in addition to the seasonal rainfall trends we observe in our datasets. Orangutan $\delta^{18}O$ values are also quite similar to the $\delta^{18}O$ values from five humans from Flores, Indonesia (14.8–21.0‰) dated at ~2.2–3.0 ka (*Vaiglova et al., 2024*). This is remarkable given the major dietary differences between frugivorous orangutans and omnivorous coastal-dwelling humans, and suggests that their enamel $\delta^{18}O$ values are predominantly influenced by regional precipitation.

## Fossil orangutan isotope values suggest different ancient climates in Sumatra and Borneo

Dating studies at Lida Ajer have established the presence of the oldest human remains in insular Southeast Asia, ~63–73 ka (*Westaway et al., 2017*), and a broad survey of the cave has reconfirmed an age of MIS 4 (59–76 ka) for the mammalian fauna (*Louys et al., 2022*). This is consistent with the minimum age of ~33–40 ka estimated for the two molars examined in the current study. The Sumatran Sibrambang Cave has been regarded as roughly contemporaneous to Lida Ajer given broad faunal similarities (*de Vos, 1983*). Recent U-series dating of two fossil orangutans from the Sibrambang assemblage yielded minimum ages of >56 ka and >85 ka (*Louys et al., 2024*), which bracket the apparent U-series minimum ages of ~60–70 ka in the current study. Sibrambang primates appear similar to, or slightly older than, those from Lida Ajer, given the minimum U-series age estimates for teeth from both sites, but this is not definitive given the absence of finite numerical ages for the fossils. Our analysis of $\delta^{18}O$ values in Sumatran orangutan fossil molars reveals a close similarity across sites and with modern Sumatran individuals, although the fossil compositions fall at the upper end of the modern range. This may indicate a slightly dryer and less variable climate during the late Pleistocene; elevated tooth $\delta^{18}O$ values are also indicative of elevated values in hydrological systems globally, resulting from increased ice volumes in glaciers and at the poles.

Pollen records from the Niah Caves archaeological site indicate that there were a number of local ecological shifts from lowland rainforest to more open environments during the Late Pleistocene and into the Holocene (*Hunt et al., 2012*), where humans may have begun hunting orangutans at ~45 ka (*Spehar et al., 2018*). While it is not possible to locate the two fossil orangutan molars in these pollen records, *Piper and Rabett, 2016*, considered that the large animal bone assemblages accumulated within the Lobang Hangus entrance and defined by the Harrisson spit depths of 12–42″ were of terminal Pleistocene age. More broadly, the orangutan specimen from grid US/22 (32–36″) is stratigraphically positioned between radiocarbon ages of 14,206–15,061 cal. BP (OxA-13936) and 36,583–38,059 cal. BP (OxA-13938), and this provides plausible minimum and maximum age constraints that are not incompatible with the apparent minimum U-series age of ~9 ka. Based on these results, the tooth is likely to date from the latest part of the Late Pleistocene. The specimen from grid Y/F4 might date from the latest part of the Late Pleistocene to the early Holocene, by comparison with the shell and fauna assemblage from other excavated areas (*Piper et al., 2016.*)

Both orangutan molars from the Niah Caves yield wide ranges of $\delta^{18}O$, which is particularly notable given the short periods of time sampled compared to the other fossils and most modern orangutan molars. Given the similar offsets in $\delta^{18}O$ values between modern baboons living in Ugandan forests and the Ethiopian rift region (*Green et al., 2022*) and modern and prehistoric Bornean orangutans, we regard the higher $\delta^{18}O$ values in the Niah Cave orangutans as possibly indicative of reduced rainfall when compared to recent conditions. This is consistent with paleoclimate reconstructions for Borneo and Flores during the late Pleistocene and early Holocene (*Griffiths et al., 2009*; *Buckingham et al., 2022*), when the environment around the Niah Caves is believed to have been a drier, more open seasonal forest (*Harrison, 1996*; *Hunt et al., 2012*). A study of $\delta^{18}O$ values in Niah Caves shell middens dating from the early to mid-Holocene indicates a shift to periods of high rainfall with less variation than modern conditions (*Stephens et al., 2016*). The transition from a drier environment to moist tropical rainforest is also reflected in the increasing number and higher frequencies of canopy-adapted mammalian taxa in excavated layers of the Pleistocene-Holocene transition (*Piper and Lim, 2021*).

Our approach has the potential to contribute to reconstructions of ancient paleoenvironments in Southeast Asia based on studies of pollen, molluscs, faunal community compositions, guano records, and stable isotopes of teeth (e.g. *Jablonski et al., 2000*; *Bird et al., 2005*; *Louys and Meijaard, 2010*; *Wurster et al., 2010*; *Hunt et al., 2012*; *Janssen et al., 2016*; *Stephens et al., 2016*; *Louys and Roberts, 2020*; *Bacon et al., 2021*; *Louys et al., 2022*; *Hamilton et al., 2024*). This may be especially timely given that recent work examining modern fauna compositions in African landscapes has cautioned that fossil herbivore assemblages tend to overestimate the extent of ancient grasslands in comparison to woodlands (*Negash and Barr, 2023*; also see *Sokolowski et al., 2023*). Fine-scaled tooth sampling may also allow an expansion of inferences from $\delta^{18}O$ values of bulk-sampled Asian hominin remains (*Janssen et al., 2016*; *Roberts et al., 2020*; *Kubat et al., 2023*), which are difficult to interpret for understanding seasonal rainfall dynamics in tropic environments (*Green et al., 2022*). Such information could better inform debates about whether humans employed arid savannah corridors to avoid dense tropical forests, or whether humans were adept at colonizing such environments during their consequential migration throughout island Southeast Asia.

## Materials and methods

### Orangutan samples

Thin (histological) sections of twelve molar teeth from six modern orangutans and six molar teeth from five fossil orangutans were employed (*Table 1*). These sections were previously prepared for studies of incremental tooth development, enamel thickness, elemental chemistry, and Asian hominoid taxonomy (*Smith, 2016*; *Smith et al., 2011*; *Smith et al., 2012*; *Smith et al., 2017*; *Smith et al., 2018b*). Four modern individuals were sourced from the Munich State Anthropological Collection (ZSM): two were collected in 1893–1894 from Skalau (north of the Kapuas River and south of the Klingkang Mountains in eastern West Borneo), and two were collected prior to 1939 from Aceh (northwest Sumatra) (*Röhrer-Ertl, 1988*: *Figure 3*, p. 14) (*Figure 1*). It was not possible to determine from which specific regions or time periods the two other modern individuals derive—collection notes were not available for these specimens from the Harvard Museum of Natural History (MCZ) or the Humboldt Museum (ZMB). Ages at death were determined for five of six individuals from assessments of incremental features and elemental registration of serially forming molars (detailed in *Smith, 2016*; *Smith et al., 2017*).

We also studied four Sumatran fossil orangutan teeth that were collected more than a century ago from the Lida Ajer and Sibrambang Caves in the Padang Highlands by Eugene Dubois (*de Vos, 1983*). Right and left lower molars from Lida Ajer (11594.12, 11595.105) show identical trace element patterns in their dentine (*Appendix 1—figure 6*), as well as similar occlusal fissure patterns and light wear, consistent with their attribution to the same individual. Two Bornean fossil orangutan teeth from Niah Caves (Malaysia) were also included in this study. The caves have yielded significant late Pleistocene and early Holocene human remains since the Harrissons began excavations in the 1950s (*Barker et al., 2007*). These lower molars were derived from two different entrances to the cave system, Gan Kira (grid square Y/F4) and Lobang Angus/Hangus (grid square US/22), with burial depths of 12–18″ and 30–36″, respectively (*Hooijer, 1961*). Although *Hooijer, 1948*, *Hooijer, 1961*, identified all six of these fossil teeth as M1s, we regard this as tentative, given that isolated orangutan molars are notoriously difficult to seriate (*Grine and Franzen, 1994*).

### Dating of fossil samples

Preliminary assessments at the Australian National University Radiocarbon Dating Laboratory confirmed that collagen preservation in the six fossil orangutans was insufficient for radiocarbon dating, a common limitation in tropical environments (e.g. *Wood et al., 2016*). Laser ablation uranium series (U-series) analyses were carried out on longitudinal sections of teeth at the Radiogenic Isotope Facility of the University of Queensland using an ASI RESOlution SE laser ablation system connected to a Nu Plasma II MC-ICP-MS. A succession of several rasters (<2 min linear ablations) was made in a transect across the dentine and enamel of each tooth (*Appendix 1—figures 3–5*) following *Grün et al., 2014*. The $^{230}Th/^{238}U$ and $^{234}U/^{238}U$ activity ratios of the samples were normalized to bracketing analyses of a homogeneous rhino tooth standard that has been precisely calibrated by isotope dilution (*Grün et al., 2014*). Importantly, dental tissues are known to behave as open systems for U-series

elements; provided there is no occurrence of uranium leaching, age estimates should therefore be regarded as minimum age constraints since uranium uptake into dental tissues may be significantly delayed after death.

## Tooth formation and oxygen isotope analyses

Thin sections were first imaged with transmitted light microscopy. Enamel daily secretion rates were measured between sequential accentuated growth lines to yield the time of formation (see *Smith, 2016*: *Figure 1*, p. 94), and enamel extension rates were calculated between accentuated lines to guide placement of the analyzed spots at approximately weekly intervals of growth from the dentine horn tip to the enamel cervix (*Smith et al., 2018a*; *Green et al., 2022*). Following the removal of cover slips by immersion in xylene, each thin section was analyzed for $\delta^{18}O$ at the SHRIMP Laboratory at the Australian National University according to methods detailed in *Vaiglova et al., 2024*.

In brief, a 15 kV Cs primary ion beam focused to a spot ~15 × 20 μm diameter was used to sequentially sample the enamel as close as possible to the EDJ. Oxygen secondary ions were extracted at 10 kV and analyzed isotopically by a multiple collector equipped with dual electrometers operated in resistor mode. The $\delta^{18}O$ values were calculated relative to reference apatite (Durango 3) measured every 10–15 sample analyses. Distances of SHRIMP $\delta^{18}O$ measurements along the innermost enamel from the cusp to cervix were converted to secretory time in days following *Green et al., 2022*. A polynomial regression relating distances to days was created using the enamel extension rates, and this regression was applied to estimate the timing of secretory deposition at every SHRIMP spot location. The Lomb-Scargle periodogram was used to assess time-dependent patterns of $\delta^{18}O$ values, which estimates the power of sine wave periods within a given range to produce the temporal patterns present within those measurements.

The probability that differences between first and second year $\delta^{18}O$ values in modern first molars might have arisen by chance was assessed by one-way paired t-tests, with alpha = 0.05 adjusted by Bonferroni correction due to repeated comparisons across multiple teeth. Figure and data plotting using Python 3 in the Google Colab environment were aided by ChatGPT, a language model based on the GPT-3.5 architecture developed by OpenAI.

## Acknowledgements

Manfred Ade, Steffen Bock, Judy Chupasko, Mark Omura, Carina Phillips, and Olav Rohrer-Ertl assisted with access to living orangutan material. Graeme Barker, Ipoi Datan, and Natasja den Ouden assisted with access to fossil orangutan material. The following museums and universities provided access to material: Humboldt Museum (Berlin), Harvard University Museum of Comparative Zoology (Cambridge), Naturalis Museum (Leiden), State Anthropological Collection (Munich), and the Sarawak Museum (Kuching). We acknowledge the Translational Research Institute for providing core facilities, particularly Kamil Sokolowski from Preclinical Imaging, as well as the Lions Club of Australia and the Mater Foundation for funding the Skyscan 1272. Kate Britton, Luca Fiorenza, Gathorne Gathorne-Hardy (Earl of Cranbrook), Terry Harrison, Jessica Rippengal, Pam Walton, and Rachel Wood provided research assistance. Sample preparation and oxygen isotope analysis was funded by the Australian Academy of Science Regional Collaborations Programme, the Australian National University, the Australian Research Council (DP210101913), and Griffith University. The U-series dating was funded by the Australian Research Council (FT150100215) and the Spanish Ramón y Cajal Fellowship (RYC2018-025221-I). The latter is funded by MCIN/AEI/10.13039/501100011033 and by 'ESF Investing in your future'. U-series dating analyses were carried out within the framework of the existing Brisbane Geochronology Alliance (Griffith University, University of Queensland, and Queensland University of Technology); Y Feng helped with the U-series data acquisition.

## Additional information

### Funding

| Funder | Grant reference number | Author |
|---|---|---|
| Australian Academy of Science | Regional Collaborations Programme | Tanya M Smith |
| Australian Research Council | DP210101913 | Tanya M Smith |
| Australian Research Council | FT150100215 | Mathieu Duval |
| Ministerio de Ciencia, Innovación y Universidades | Ramón y Cajal Scholarship RYC2018-025221- I | Mathieu Duval |

The funders had no role in study design, data collection and interpretation, or the decision to submit the work for publication.

### Author contributions

Tanya M Smith, Conceptualization, Data curation, Formal analysis, Funding acquisition, Investigation, Methodology, Writing - original draft, Project administration, Writing – review and editing; Manish Arora, Resources, Funding acquisition; Christine Austin, Resources, Visualization; Janaína Nunes Ávila, Formal analysis, Validation, Writing – review and editing; Mathieu Duval, Data curation, Formal analysis, Visualization, Writing – review and editing; Tze Tshen Lim, Resources, Writing – review and editing; Philip J Piper, John de Vos, Validation, Writing – review and editing; Petra Vaiglova, Formal analysis, Visualization; Ian S Williams, Resources, Formal analysis, Funding acquisition, Validation, Methodology, Writing – review and editing; Jian-xin Zhao, Resources, Funding acquisition, Validation, Writing – review and editing; Daniel R Green, Software, Formal analysis, Funding acquisition, Validation, Investigation, Visualization, Methodology, Writing – review and editing

### Author ORCIDs

Tanya M Smith [ID] http://orcid.org/0000-0001-8175-8208
Janaína Nunes Ávila [ID] http://orcid.org/0000-0003-0035-6309
Petra Vaiglova [ID] http://orcid.org/0000-0002-9468-8138

Reviewer #1 (Public Review): https://doi.org/10.7554/eLife.90217.3.sa1
Reviewer #2 (Public Review): https://doi.org/10.7554/eLife.90217.3.sa2
Author Response https://doi.org/10.7554/eLife.90217.3.sa3

## Additional files

### Supplementary files

• Supplementary file 1. U-series dates for fossil orangutan material. $^{230}$Th/$^{238}$U and $^{234}$U/$^{238}$U are activity ratios. It is worth noting that, for most transect analyses, the $^{232}$Th signal, which was measured on a Faraday collector, was indistinguishable from background noise. In this regard, the corresponding $^{230}$Th/$^{232}$Th activity ratio of each transect should be >>100, and thus non-radiogenic or detrital $^{230}$Th correction would have negligible impact on the age. U-series data in italics should be viewed with caution due to U concentrations of ≤0.5 ppm. All errors are 2-σ. Key: EN = enamel; DE = dentine; n.c.=not calculable. Negative values were caused by background extraction from their measured peaks with intensities at detection levels.

• MDAR checklist

• Source data 1. Source data for *Figures 2–4*, *Appendix 1—figures 1 and 2*.

### Data availability

All data generated or analysed during this study are included in the manuscript and supporting files; source data files have been provided for Figures 2–4, Figure 4—figure supplement 1, and Appendix 1—figure 1.

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

## Appendix 1

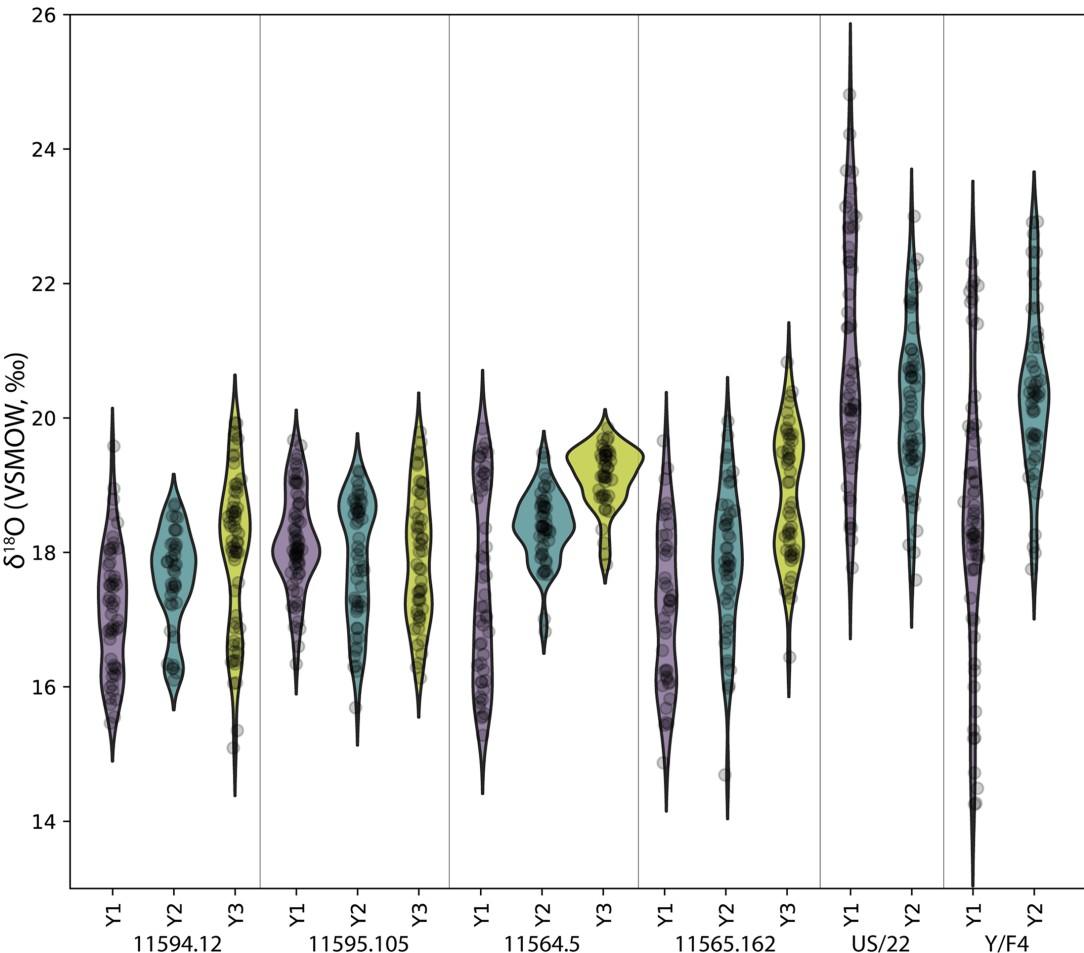

**Appendix 1—figure 1.** Comparison of sequential δ$^{18}$O values across multiple years of molar formation in six putative fossil orangutan M1s from Borneo and Sumatra. The width of each curve is a kernel density estimate (KDE) corresponding to the distribution of oxygen isotope values measured from different teeth. From each tooth, first year data (Y1) is shown with a purple violin plot, second year data (Y2) with a green plot, and third year data (Y3, if present) with a yellow plot. Actual data are plotted as black circles.

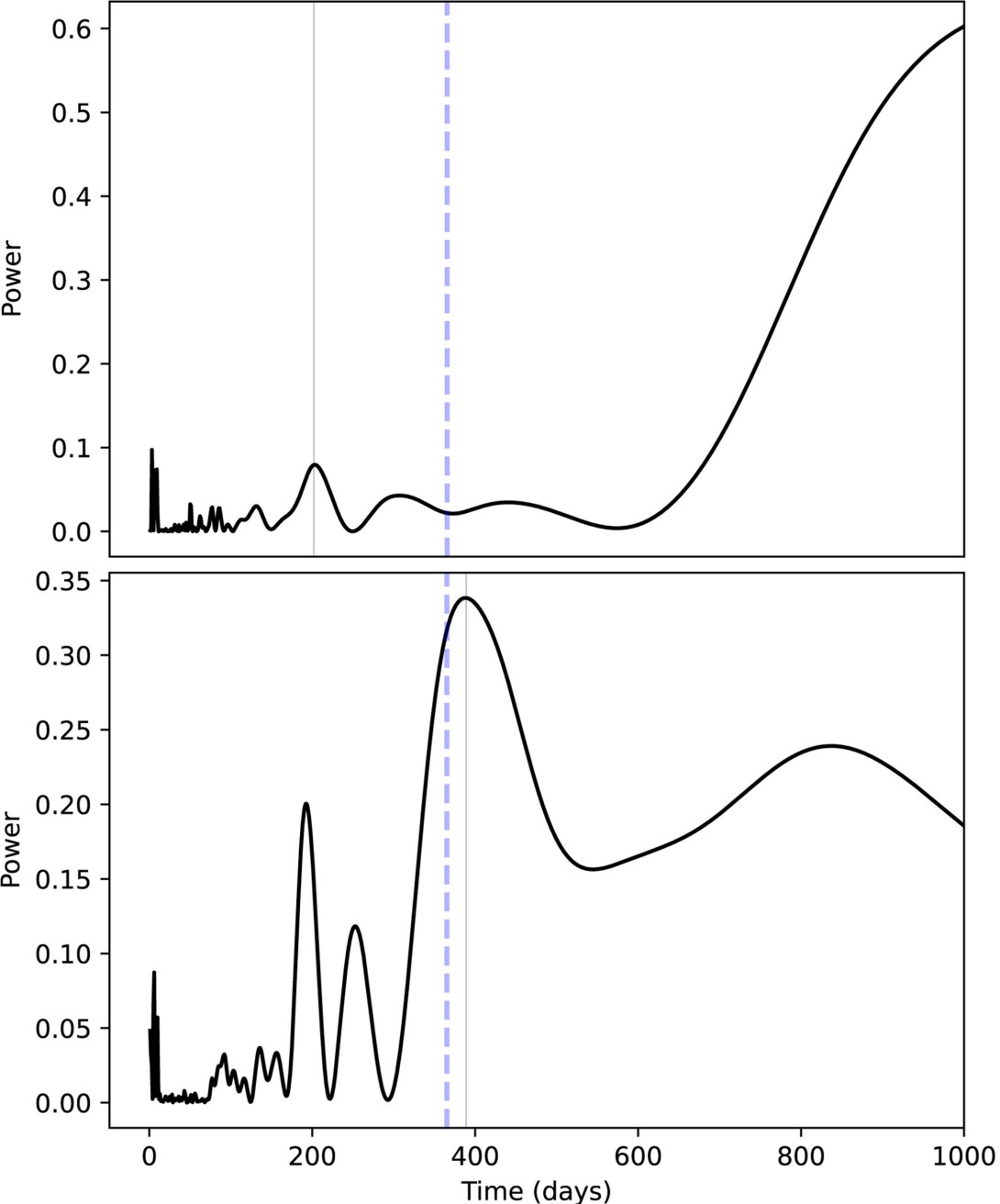

**Appendix 1—figure 2.** Inferred seasonality of δ¹⁸O values from a Bornean (top) and Sumatran (bottom) M1.
Lomb-Scargle periodograms show potential periods in days (x-axis) against period power (y-axis), where higher
values on the y-axis indicate underlying sine wave periods that produce, contribute to, or explain δ¹⁸O value
oscillations. Best-fit periodicities are shown as light gray vertical lines, whereas annual periodicities are indicated by
blue dashed horizontal lines. We regard the Bornean M1 (MCZ 5290) as largely aperiodic; a minor peak is observed
at c. 6 months, and increasing powers at very high periods are an artifact of limited sampling length within teeth
relative to the model. The Sumatran M1 (ZMB 83508) has a 1.1 year inferred frequency, as well as an approximately
6-month peak, likely reflecting semiannual monsoons.

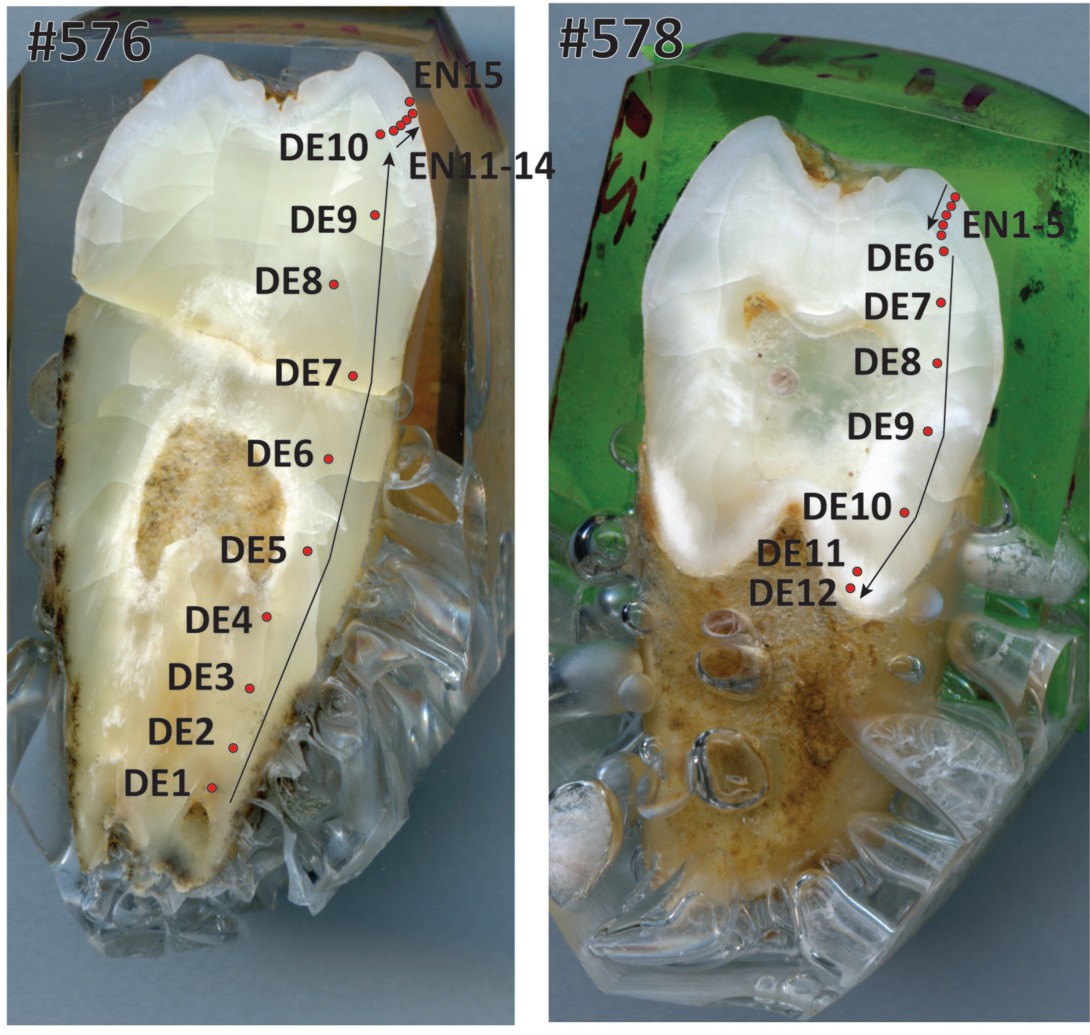

**Appendix 1—figure 3.** Laser ablation profiles performed across the two teeth from Lida Ajer. #576 (left) refers to specimen 11595.105; #578 (right) refers to specimen 11594.12. The red dots represent the position of the rasters, and arrows indicate the sequence of the analyses. EN = enamel. DE = dentine.

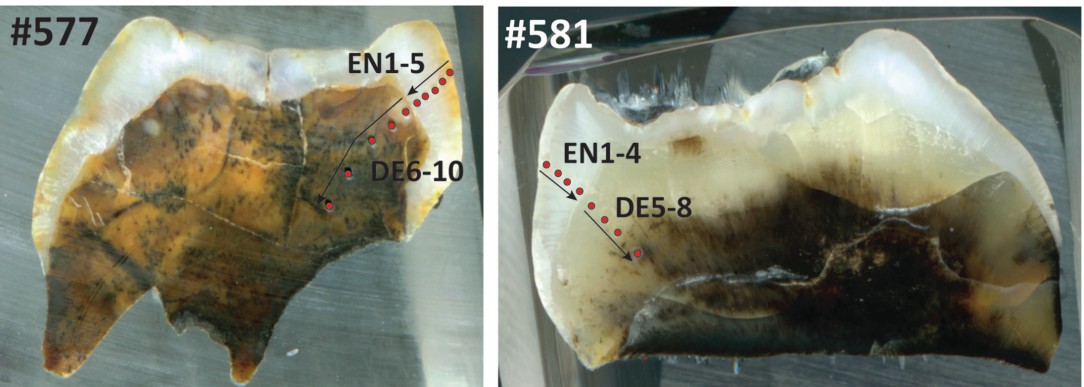

**Appendix 1—figure 4.** Laser ablation profiles across the two teeth from Sibrambang Cave. #577 (left) refers to specimen 11565.162; #581 (right) refers to specimen 11564.5. The red dots represent the position of the rasters, and arrows indicate the sequence of the analyses. EN = enamel. DE = dentine.

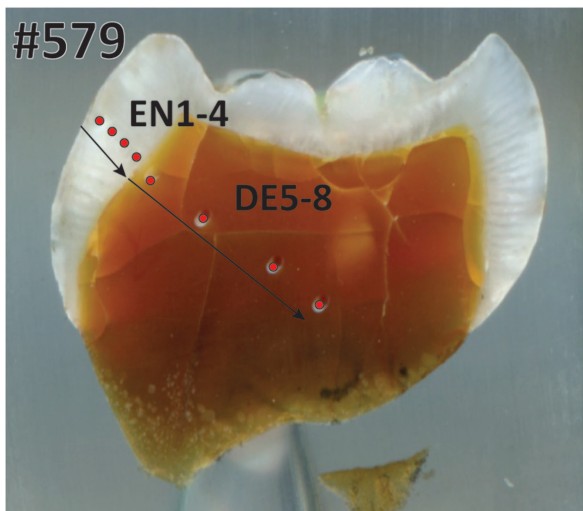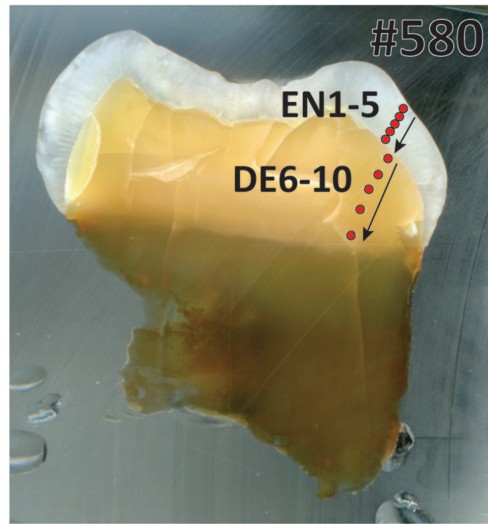

**Appendix 1—figure 5.** Laser ablation profiles across the two teeth from Niah Caves. #579 (left) refers to specimen from grid Y/F4; #580 (right) refers to specimen from grid US/22. The red dots represent the position of the rasters, and arrows indicate the sequence of the analyses. EN = enamel. DE = dentine.

11594.12

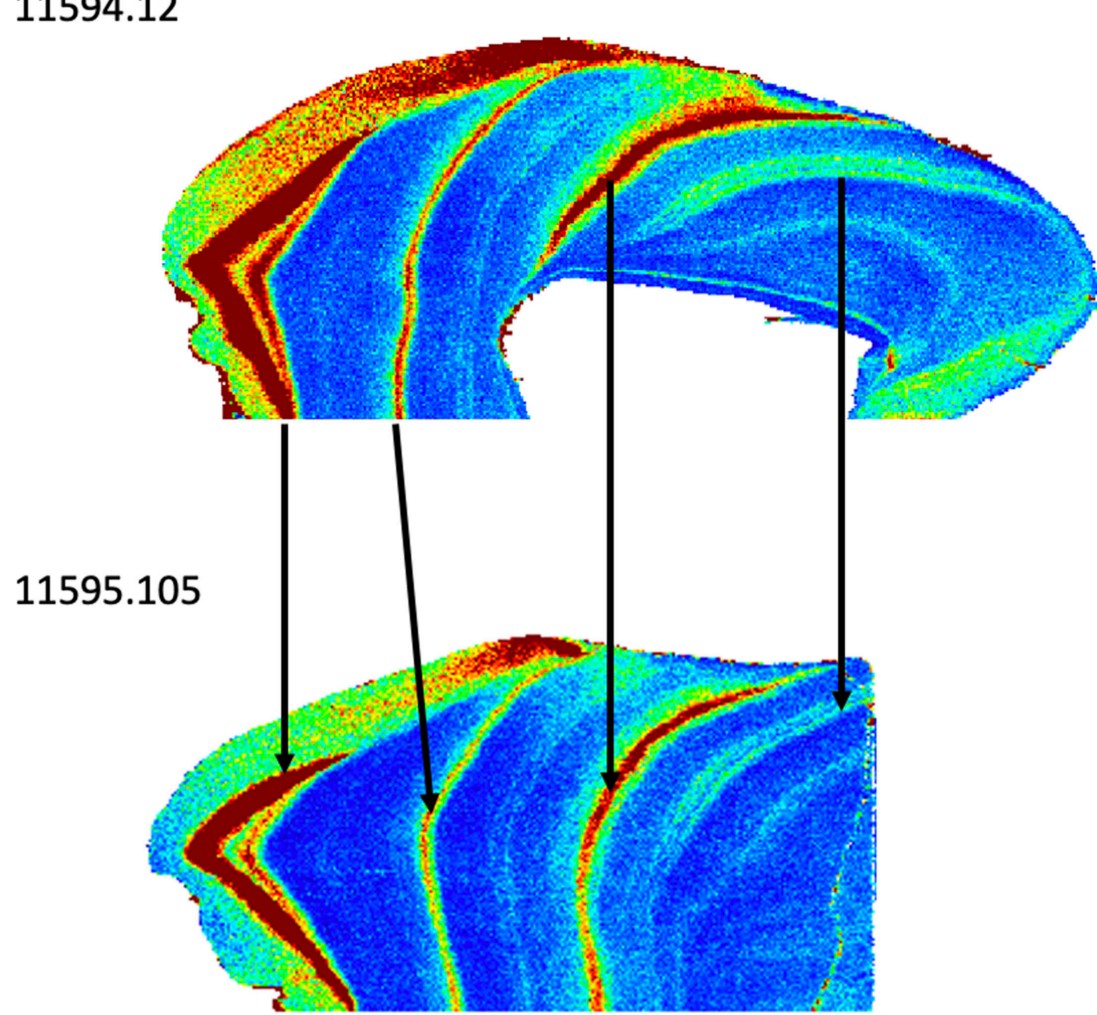

11595.105

**Appendix 1—figure 6.** Matching trace element patterns in cross-sections of two isolated molars from the Dubois collection of fossil orangutan teeth from Lida Ajer. High concentrations are shown in warm colors, low concentration are in cool colors; here Li/Ca is shown, but identical corresponding patterns were also observed for Ba/Ca and Sr/Ca (not shown). The enamel cap of each tooth is to the left, and root dentine is to the right. Trace elements were measured according to LA-ICP-MS methods detailed in *Smith et al., 2017*.

