## [Editor Report · eLife assessment]

This **important** study presents **convincing** evidence for the use of orangutan teeth as terrestrial proxies to reconstruct rainfall regimes, while exploring the potentially conflicting impact of breastfeeding signals. The findings will be of broad interest for those using and developing methods and tools to reconstruct environmental conditions in the historical and archaeological past.

---

## [Referee Report · Reviewer #1 (Public Review)]

Summary:

The authors measured the oxygen stable isotope ratios in six orangutan teeth using a state of the art micro-sampling technique (SHRIMP SI) to gather substantial multi-year isotopic data for six modern and five fossil orangutan individuals from Borneo and Sumatra. This fine-scale sampling technique allowed to address the fundamental question if breastfeeding affects the oxygen isotope ratios in teeth forming in the first one to two years of life, during which orangutans can be assumed to largely depend on breastmilk. The authors provide compelling evidence that the consumption of milk does not appear to affect the overall isotopic profile in early forming teeth. They conclude that this allows us to use these teeth as terrestrial/arboreal isotopic proxies in paleoenvironmental research, which would provide an invaluable addition to otherwise largely marine climate records in this regions.

Strengths:

The overall large sample size of orangutan dental isotope records as well as the rigorous dating of the fossil specimens provide a strong dataset for addressing the outlined questions. The direct comparison of modern and fossil orangutan specimens provides a valuable evaluation of the use of these modern and past environmental proxies, with some discussion of the implications for the environmental conditions during the expansion of early modern humans into this region of the world.

Weakness:

The authors illustrate that all orangutan individuals sampled, modern and fossil, show a considerable amount of isotopic variation between and within their teeth. Some of this variation is clearly associated with isotopic shifts in precipitation, but some will also be linked to the variation in oxygen isotopes within the forest itself and the many plant foods it produces for the orangutan. In the future, the systematic measurement of oxygen isotopes across orangutan food items, forest canopies and precipitation could help differentiate how much of the observed isotopic variation in teeth is indeed related to climatic shifts alone.

---

## [Referee Report · Reviewer #2 (Public Review)]

Summary:

This manuscript provides microprobe serial oxygen isotope data from thin-sectioned modern and fossil orangutan teeth in an effort to reconstruct seasonality of rainfall in Borneo and Sumatra. The authors also explore the hypothesis that nursing could affect early tooth (first molar) isotope values. They find that all molars yield similar oxygen isotope values and therefore conclude that future research need not exclude use of first molars. With regard to seasonality, the modern orangutans yield similar results from both islands. The authors suggest differences between modern and fossil orangutan teeth.

Strengths:

The study employs a sampling method that captures serial isotope values within thin sections of teeth using a microprobe that provides much higher resolution than traditional hand-held drilling.

Weaknesses:

The study only examines six modern and six fossil orangutan individuals. Of those, only four modern individuals were samples across multiple molars.

---

## [Author Response]

The following is the authors’ response to the original reviews.

**Reviewer #1 (Recommendations For The Authors):**
I highly appreciate this study and found the paper to be very well-written and easy to follow. However, a more extensive discussion of what I summarized under "weakness" would strengthen the paper. This may include a broader discussion of the canopy effect itself and the most relevant literature on its extent in rainforest settings in general and primate foods in particular, as well as more details on the dietary behavior of modern orangutans (stratigraphy of orangutan foods) and how seasonal their diet is. The extreme seasonality in orangutan plant food availability should be discussed. Now there are only 2 sentences in the discussion (lines 304-312) and I find the word "plant' only twice overall, though variation in plant food d18O is what drives variation in orangutan dental d18O values.

We very much appreciate the support of this reviewer, and their feedback about the clarity of the paper. As noted in the provisional reply to reviewers, we are happy to add additional context about the issue of isotopic enrichment within forest canopies, and have expanded the original paragraph in the discussion devoted to this subject. We made reference to the fact that orangutan diets vary by season and site in the original submission, and have now acknowledged that seasonal diet variation may also contribute to variation in enamel isotope values.

Also, I'd like to note that there has been only one recent study so far that made some level of an attempt to find a breastfeeding effect in orangutans using fecal isotope data. Tsutaya et al. 2022 (AJBA) report some seasonality in adult orangutan fecal isotope values, which could be relevant here as well. But also they reported some data from 2 to 7-year-old orangutan offspring and did not see any breastfeeding pattern in isotope values here either. Probably not too surprising at this older age, but still worth noting in the context of this study.

There is a 2019 study that sampled fecal isotopes in 43 mother-infant orangutan pairs and found a different pattern than Tsutaya et al. (2022), although these data have not been published in full (Knott et al. (2019) AJBA 168, S68, 128-129). Given these contradictions, the fact that neither study serially sampled the first two years of life, and caveats to fecal isotope sampling of wild primates reviewed in Bădescu et al. (2023: American Journal of Primatology 2023;e235), introducing these nitrogen isotope studies does not aide in the interpretation of oxygen isotope data during intensive nursing, and thus is beyond the scope of this paper. The seasonality Tsutaya et al. (2022) reported in adult fecal samples was for carbon isotopes rather than nitrogen isotopes, and its relevance to the current study is unclear given that the orangutan plant foods measured did not show seasonal variation in carbon isotopes. As requested above, we have noted orangutans’ dietary seasonality might influence the variation of oxygen isotope values.

**Reviewer #2 (Recommendations For The Authors):**
First, the manuscript offers upfront flashy numbers with respect to the number of samples, but what the reader really needs to know upfront is the number of individuals and the number of teeth per individual. These facts are buried and make the reader work too hard to keep track. While the specimen ID numbers are valuable in the table, perhaps a different ID could be used in the text, such as individuals modern Borneo A and B, fossil Sumatra A and B, etc.? Similarly, it would be helpful to remind readers of each locality - Borneo or Sumatra, modern or fossil.

Tables 1 and 2 and the first sentence of the results and the materials and methods stated that we measured 18 teeth in this study. It is likely that the placement of the tables at the very end of the manuscript in the submitted version made the sample sizes and specimen information less evident to the reviewer. In response to this critique we have now added the number of teeth to the abstract, and trust that when the tables are placed within the text as indicated it will be easier to follow textual references to particular individuals. Museum identification codes have been provided in two previous publications of these teeth, and we retain them here for consistency.

Second, the manuscript mentions some climate change in Sumatra, but what about Borneo?

The results on the Bornean fossil teeth stated: “The range of values from these two fossil molars (14.2–24.8 ‰) markedly exceeds the range of modern Bornean orangutans (12.7–20.0 ‰) (Figure 4), with the mean δ18O value at least 2‰ heavier, suggesting possibly drier conditions with greater seasonality during their formation.” In the final section of the discussion, we devoted two paragraphs to discussing evidence for climate change at Niah Cave in Borneo - more than we devote to discussing such data from Sumatra.

The most valuable figure in the manuscript is Figure 3 showing the serial sampling of modern teeth. It would be incredibly useful to see a similar graph for the fossils and a graph of the modern and fossils together for each island. The violin plots demonstrate a range of values but fail to provide the important seasonality signals. The manuscript is promising but as written is difficult to follow, and the results and conclusions with regard to climate change need more demonstration. On a minor note, I found myself wanting to know about the dates of fossils before knowing the isotopic values. You might wish to move the dating section to precede the isotopes.

As requested, we have added an additional Supplemental figure making the comparisons of seasonality between fossil and modern individual more evident.